# Biosolids Benefit Yield and Nitrogen Uptake in Winter Cereals without Excess Risk of N Leaching

Silvia Pampana *, Alessandro Rossi and Iduna Arduini

Department of Agriculture, Food and Environment, University of Pisa, 56124 Pisa, Italy; alessandro.rossi@agr.unipi.it (A.R.); iduna.arduini@unipi.it (I.A.)
* Correspondence: silvia.pampana@unipi.it; Tel.: +39-050-2218941

**Abstract:** Winter cereals are excellent candidates for biosolid application because their nitrogen (N) requirement is high, they are broadly cultivated, and their deep root system efficiently takes up mineral N. However, potential N leaching from BS application can occur in Mediterranean soils. A two-year study was conducted to determine how biosolids affect biomass and grain yield as well as N uptake and N leaching in barley (*Hordeum vulgare* L.), common wheat (*Triticum aestivum* L.), durum wheat (*Triticum turgidum* L. var. *durum*), and oat (*Avena byzantina* C. Koch). Cereals were fertilized at rates of 5, 10, and 15 Mg ha$^{-1}$ dry weight (called B5, B10, and B15, respectively) of biosolids (BS). Mineral-fertilized (MF) and unfertilized (C) controls were included. Overall, results highlight that BS are valuable fertilizers for winter cereals as these showed higher yields with BS as compared to control. Nevertheless, whether 5 Mg ha$^{-1}$ of biosolids could replace mineral fertilization still depended on the particular cereal due to the different yield physiology of the crops. Moreover, nitrate leaching from B5 was comparable to MF, and B15 increased the risk by less than 30 N-NO$_3$ kg ha$^{-1}$. We therefore concluded that with specific rate settings, biosolid application can sustain yields of winter cereals without significant additional N leaching as compared to MF.

**Keywords:** barley; biosolids; common wheat; durum wheat; mineralization; N leaching; N uptake; nitrates; oat



## 1. Introduction

The growing global population has concurrently increased food demand and organic waste production, with the former requiring increasing areas of arable land for crop production, and the latter subtracting it for waste disposal. On the one hand, the reuse of organic wastes through land application would reduce the land surface subtracted for disposal; on the other hand, it would enhance the recovery of resources, fully complementing the concepts of the Circular Economy Package adopted by the European Commission [1].

Sewage sludge is a semi-solid organic waste resulting from the wastewater treatment process, and it is termed "biosolids" (BS) when treated to reach the appropriate standard set by regulations for land disposal [2].

The agricultural benefits of BS utilization in agriculture are well-documented and mainly attributed to low-cost access to nutrients and organic matter [3–5]. In particular, their high nitrogen (N) and phosphorous (P) content makes BS attractive as a fertilizer source [6]. Biosolids may have some advantages over inorganic fertilizers, the most important being that they can better match the nitrogen demand of crops. This is because the organically bound N, which is the prevailing chemical form in BS, is more stable compared to the N forms in mineral fertilizers, and therefore less soluble in water; likewise, it must be mineralized into plant available nitrogen (PAN) before it can be assimilated by crops [6].

Despite these advantages, the use of BS in agriculture is not widely accepted, primarily because of the possible environmental impacts. In the last decades, however, firmer regulations and improved pre-treatment technology and monitoring of pathogens and

pollutants have resulted in reduced pathogen burdens and decreased concentrations of heavy metals and organic contaminants [7]. Thus, at present, the addition of high-quality BS raises environmental concerns predominantly regarding the risks of the excess release of nitrate in soil and water [8,9].

In the Mediterranean region, the non-point-source nitrate pollution of aquifers is regarded as one of the main environmental impacts of agriculture because of its unique climate characteristics and agricultural peculiarities [10].

Winter cereals are excellent candidates for BS application because: (i) they rapidly form a dense and deep root system that can efficiently take up mineral N; (ii) their N requirement is high; (iii) they are cultivated in vast areas of Mediterranean countries.

Nevertheless, biosolid application before the sowing of winter cereals could pose a high risk of N leaching as rainfall is concentrated between November and February [11].

The application of BS has been proved to be an agronomically and environmentally suited practice in various wheat cropping systems [12–14]. In durum wheat, both [15] and [16] found that the application of sewage sludge appeared to be more beneficial than mineral fertilization; biosolids were also an effective substitute to mineral fertilizers as a source of N in barley [11], while [17] reported that applying BS significantly increased straw but not the grain yield. Similarly, oat plants amended with BS improved their biomass production by about 30% as compared to conventional ammonium sulphate [18].

However, inconsistences in literature findings highlight that the establishment of sustainable management systems still requires site-specific evaluation of both the BS fertilization value and the associated N-leaching risks.

In field experiments, crop biomass and N uptake at different stages as well as N leaching during the crop cycle could be used as indicators of the rate of organic N mineralization [19]. However, results can greatly vary depending on biosolid properties and site characteristics, which encompasses both soils and climate.

The optimal rates of biosolid application are greatly dependent on the nature of biosolids, on crop demand, and pedo-climatic characteristics, and should be tailored to specific cropping systems so as to meet crop requirements and minimize environmental impact [13].

In Italy, the regulation of BS application in agriculture (Dlgs 99/1992) sets a limit of 15 Mg ha$^{-1}$ of dry matter (DM) applied to arable soils, with $6 < pH < 7.5$ and CEC $> 15$ meq 100 g$^{-1}$ over a period of three years [7]. This amount could be applied once with a single application in three years, or three or two times with split applications at accordingly lower rates.

The one-time application of BS at a high rate may greatly differ from the repeated annual applications at lower rates, both in supplying N to crops and in influencing the risk of N leaching. However, little experimental data are available on the effects of different application rates on diverse crops in determining grain yield and N leaching.

To fill this gap, the present research aimed to investigate the effects of BS applied at three different rates on the growth, grain yield, and N uptake, as well as on N losses via leaching, in four winter cereals: barley, common wheat, durum wheat, and oat.

The fertilization value and the N-leaching potential of biosolids were compared with those of unfertilized and mineral-N fertilized controls.

## 2. Materials and Methods

### 2.1. Site Characteristics and Experimental Design

The research was carried out for two cropping seasons (harvesting years 2015 and 2017) in an open-air, semi-controlled facility, at the Research Centre of the Department of Agriculture, Food, and Environment of the University of Pisa, Italy, which is located approximately 4 km from the sea (43°40 N, 10°19 E) and 1 m above sea level.

The climate of the area is Mediterranean (Csa), according to Köppen classification, with low temperatures in winter that increase rapidly during spring, a hot summer, and an irregular pattern of yearly rainfall distribution, generally concentrated in autumn

and spring. The long-term mean annual maximum and minimum daily air temperature of the site are 20.2 °C and 9.5 °C, and the mean annual rainfall is 971 mm.

The experiment was carried out in pots of four winter cereals, arranged in a randomized block design, using five fertilization strategies, and replicated four times in both years. For each crop, the most widely used cultivar in Central Italy were selected: two-row barley (*Hordeum distichon* L., cv Scarlett), common wheat (*Triticum aestivum* L., cv Bologna), durum wheat (*Triticum turgidum* subsp. *durum* (Desf.) Husn., cv Claudio), and oat (*Avena sativa* subsp. *byzantina* (K. Koch) Romero Zarco, cv Argentina).

The five fertilization strategies were:

- Unfertilized control (C);
- Standard mineral fertilization at the rate of 150 kg N ha$^{-1}$ (MF) applied in three fractions: 30, 60, and 60 kg N ha$^{-1}$, respectively, (i) at sowing with ammonium sulphate, (ii) at the beginning of stem elongation (BBCH 30), and (iii) at first node 1 cm above tillering node (BBCH 31), with urea;
- Biosolids at three different rates: 5 (B5), 10 (B10), and 15 (B15) Mg DM ha$^{-1}$ applied 7 days prior to sowing.

Anaerobically digested and dewatered BS were obtained from the wastewater treatment plant of Livorno (Italy). The characteristics of these materials did not differ in the two years, and they had also been employed in our previous experiments [11,20]; selected properties of BS are reported in Table 1.

**Table 1.** Selected properties of BS (DM basis) applied to barley, common wheat, durum wheat, and oat in 2015 and 2017.

| Parameter | u.m. | Value |
|---|---|---|
| Moisture | % | 87 |
| pH | | 6.5 |
| Total organic C | % | 38.5 |
| Total N | % | 7.9 |
| Total P | % | 1.2 |
| Humification degree | | 1.9 |
| Total phenolic compounds | g kg$^{-1}$ | 0.6 |
| CrVI | mg kg$^{-1}$ | <1 |
| As | mg kg$^{-1}$ | <5.0 |
| Cd | mg kg$^{-1}$ | <2.0 |
| CrIII | mg kg$^{-1}$ | 16 |
| Hg | mg kg$^{-1}$ | <0.1 |
| Ni | mg kg$^{-1}$ | 25 |
| Pb | mg kg$^{-1}$ | 12.5 |
| Cu | mg kg$^{-1}$ | 72.4 |
| Zn | mg kg$^{-1}$ | 185.1 |

The composition of the studied BS complied with Italian regulations for the agricultural utilization of sewage sludge, which sets the minimum values for organic matter (>20% of DM), total nitrogen (>1.5% of DM), and total phosphorus (>0.4% of DM) as well as all the requirements (with respect to Directive 86/278/CEE) for heavy metals, pathogens, and organic micropollutants, both in BS and soils.

The rate of mineral N supply was the recommended value for optimal cereal production in Central Italy, and the adopted splitting management was proved to be the best mineral fertilization practice in the Mediterranean climate to ensure both production quantity and quality [21].

Phosphorus and potassium fertilizers were applied at sowing at the rates of 150 kg ha$^{-1}$ of P$_2$O$_5$ and K$_2$O as triple superphosphate and potassium sulphate, respectively.

## 2.2. Experimental Equipment and Crop Management

The experiment was carried out in cylindrical pots 16 cm in diameter and 60 cm in length, made of polyvinyl chloride, spaced 5 cm apart, and embedded in expanded clay to avoid daily fluctuations in soil temperature. The bottom layer of each pot was fitted with a fine cloth mesh that retained the substrate but allowed for water movement, with a valve connected at the base to permit leachate collection.

To simulate field conditions, the pot-culture experiments were carried out under open-air conditions.

In both growing seasons, pots were filled with soil collected from a field previously cultivated with rapeseed (*Brassica napus* L.), just before seeding. Soil properties did not differ between the two years and were approximately: 78.6% sand (2 mm < Ø < 0.05 mm); 18.5% silt (0.05 mm < Ø < 0.002 mm); 2.9% clay (<0.002 mm); 8.4 pH; 1.0% organic matter (Walkley and Black method); 0.5 g kg$^{-1}$ total N (Kjeldahl method); 5.4 mg kg$^{-1}$ $P_2O_5$ available (Olsen method); 807.7 mg kg$^{-1}$ $K_2O$ exchangeable (Dirks-Sheffer method); and 11.7 meq 100 g$^{-1}$ ($BaCl_2$-TEA method). BS, PK fertilizers, and the first fraction of MF treatment were mixed with the soil using a concrete mixer before pot filling, whereas second and third MF fractions were top-dressed.

Sowing was manually performed on 7 January 2015 and on 16 December 2016, which are within the optimum sowing time for winter cereal production in Central Italy. Eight viable seeds for each pot, corresponding to 400 plants m$^{-2}$, were sown.

In both years, all crops were irrigated from flowering to maturity to prevent drought stress. In this period, 150 mm of irrigation water were applied in both years. Weed control was performed throughout the two cropping seasons by hand hoeing, and plants were free from insects and diseases.

## 2.3. Sampling Procedures and Measurements

An automatic meteorological station located at the experimental site recorded rainfall, and minimum and maximum air temperatures daily.

Throughout the experiment, phenological phases were recorded using the BBCH scale for cereals [22] to determine the timing of inorganic N application and harvest.

At maturity (BBCH 99—28 June 2015 and 19 June 2017), plants were manually cut at the ground level and the number of plants, culms, and spikes per pot was recorded. Then, shoots were separated into culms + leaves and spikes. Dead leaves were also collected.

Dry weight (DW) of above ground parts (VAP) were determined and spikes (wheat and barley) or panicles (oat) (hereinafter called heads) were counted and separated into kernels and chaff. The total amount and the number of aborted spikelets per head were additionally counted. Mean kernel weight was measured by counting and weighing kernels. Head fertility index was determined as the ratio between grain number and chaff dry weight [23]. Harvest Index (HI) was calculated as the ratio of grain yield to total aboveground biomass at maturity.

After shoot removal, roots were recovered from the soil by gently washing with low-flow sprinklers. Dry weights of all plant parts were determined by oven-drying at 65 °C to a constant weight. All plant parts were analyzed for N concentration using the micro-Kjeldahl standard method. The N content was obtained by multiplying N concentrations of different plant parts by DW. Nitrogen Harvest Index (NHI) was obtained as the ratio of N content in grains to the aboveground N content.

In the second season (2017), crops were additionally sampled at flowering (BBCH 65) to better define patterns of N uptake during the crop cycle; thus, aboveground and root dry weights were determined, together with their N concentration and content.

Accordingly, in 2015, 80 pots in total were arranged (4 crops × 5 fertilizer treatments × 4 replications), each one containing eight plants and representing an experimental unit; in 2017 the number of pots was duplicated to permit additional sampling at flowering.

### 2.3.1. Leachates

To quantify N leaching, a drainage-sampling device was installed at the bottom of each pot by connecting a rigid PVC drain 3 cm in diameter to a 25 dm$^3$ PVC tank.

In both years, leachates from each container were collected after each rainfall event that produced percolation, during the entire research period. Leachate volumes were measured, and their NO$_3$-N concentration was determined through ion chromatography using DX-100 Dionex (Sunnyvale, USA). The amount of leached NO$_3$-N was calculated by multiplying the N concentration per volume of percolated water.

### 2.3.2. Statistical Analysis

A two-way ANOVA, with year and fertilization as treatments, was carried out for DW of VAP and roots, grain yield, and yield components, together with N concentration and content at maturity in the two years, separately for each species.

Moreover, dry weights of vegetative aboveground plant parts and roots, N concentration and N content at flowering in the second season, as affected by the fertilization treatment, were analyzed by one-way ANOVA.

Drainage water, NO$_3$-N concentration, and N leached from each percolation event and the cumulative amounts, as affected by the fertilization treatment, were subjected to one-way ANOVA.

The normality of data was checked using Shapiro–Wilk tests, and homogeneity of variances was tested through Levene's tests, prior to analyses.

Significantly different means were separated at the 0.05 probability level by the Tukey test.

## 3. Results

ANOVA revealed significant differences between years and among fertilization treatments; however, year x fertilization interaction was not significant for any of the measured parameters. Thus, the significant main effects of year and fertilization are presented, with results consistently averaged over fertilizations or years, respectively.

### 3.1. Weather Conditions

The thermal and humidity conditions in the two seasons were dissimilar, and better conditions for cereal growth and development were recorded in the first year (2015). Total rainfall over the two growing seasons was 316 mm in 2015 and 247 mm in 2017, respectively 68% and 75% lower than the long-term average. A significant portion of rainfall occurred in January–February in 2015 and in February–March in 2017 (Figure 1).

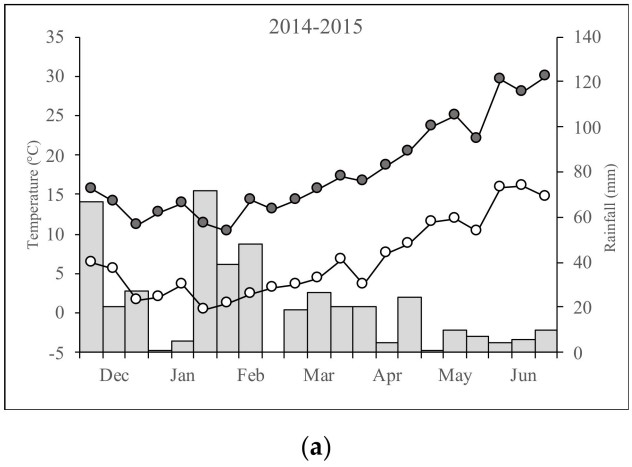
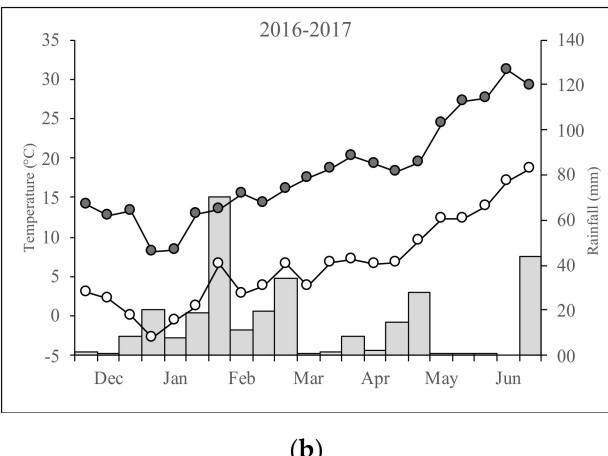

(**a**)  (**b**)

**Figure 1.** Air minimum (white dots) and maximum (black dots) temperatures and rainfall (bars) in the two cropping seasons: (**a**) 2014–2015; (**b**) 2016–2017.

Average temperatures ranged from −3.4 °C to 34.5 °C in the first growing season and from −6.3 °C to 33.4 °C in the second one; both did not differ from the 25-year average.

The lowest monthly average minimum temperature occurred in January in both years, at 0.5 °C in 2015 and −0.3 °C in 2017; maximum temperatures were recorded in June and were 29.9 and 31 °C, respectively.

### 3.2. Year Effect on Crop Growth

Differences in environmental conditions prompted significant variations in biomass production between the two years, as revealed by ANOVA.

In the first season, all the four cereals had increased VAP dry weights by 48, 55, 62, and 66%, respectively for durum, common wheat, oat, and barley (Figure 2). Root biomass was significantly higher (+64%) only in barley, being 1.5 g plant$^{-1}$ in 2015 and 0.9 in 2017; however, root biomass in common and durum wheat as well as in oat did not differ between years.

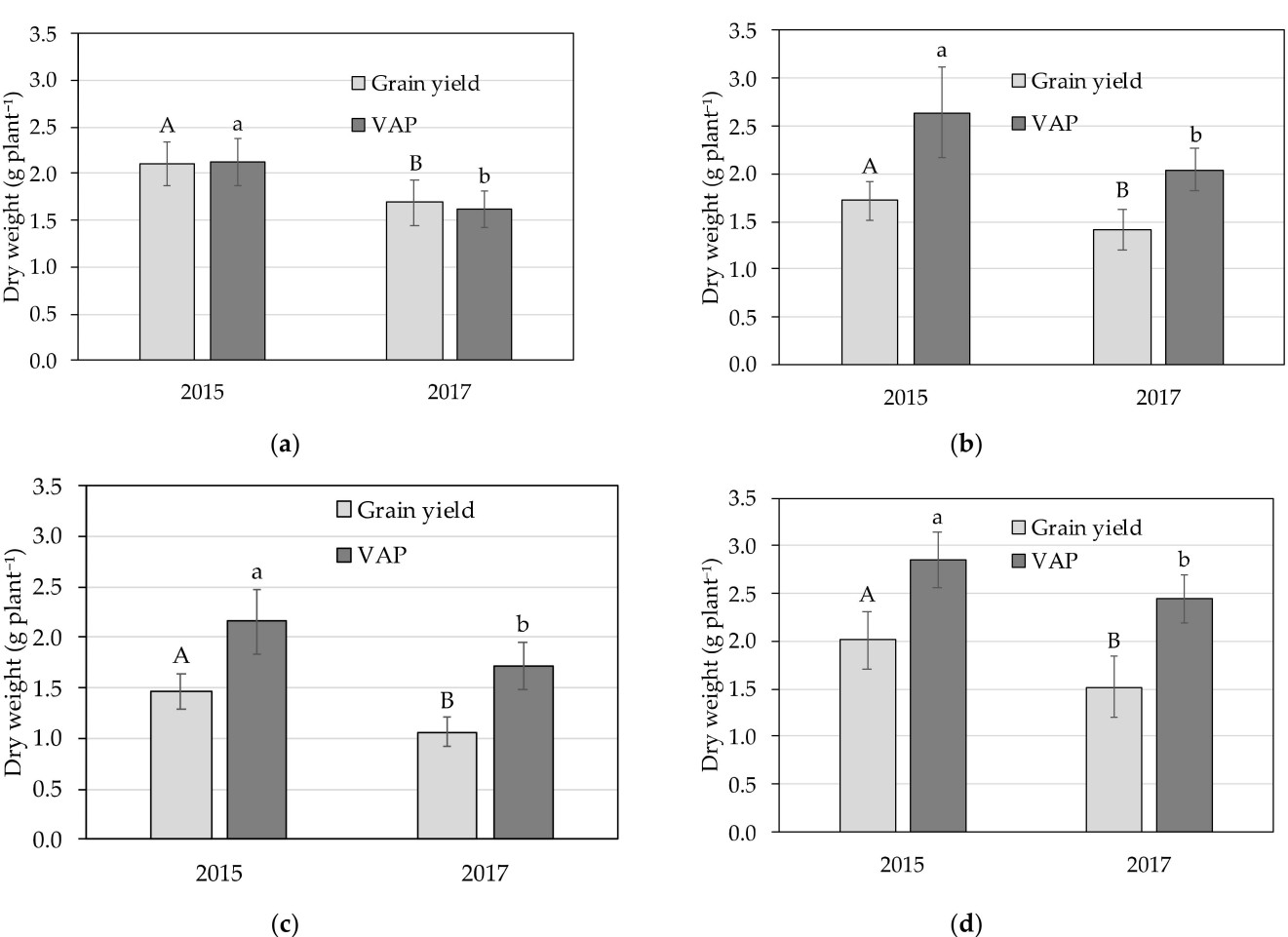

**Figure 2.** Biomass (DW g plant$^{-1}$) of vegetative aboveground plant parts (VAP) and grain yield, as affected by year treatment in: (**a**) barley; (**b**) common wheat; (**c**) durum wheat; and (**d**) oat. Bars with different letters are significantly different according to Tukey's test ($p > 0.95$); uppercase letters refer to grain yield and lowercase to VAP. Error bars represent standard deviation.

In all crops, the N concentration of VAP and roots was not affected by the year; thus, the N content followed a pattern similar to that of dry weight (Supplementary Table).

Likewise, ANOVA showed significant differences between years in grain yield, which was superior in 2015 to that in 2017 by 20% in barley, 40% in common wheat and oat, and 50% in durum wheat (Figure 2).

In all the crops, the higher grain yield of the first year was as the number of kernels per plant was augmented in barley by 16%, in common wheat and oat by 44%, and by up to 53% in durum wheat (Table 2).

**Table 2.** Yield components as affected by year treatment in barley, common wheat, durum wheat, and oat.

| Crop | Year | Heads | | Kernels | | MKW | |
|---|---|---|---|---|---|---|---|
| | | n plant$^{-1}$ | | n head$^{-1}$ | | mg | |
| Barley | 2015 | 2.5 | a | 38.5 | a | 40.4 | ns |
| | 2017 | 1.4 | b | 19.5 | b | 39.9 | ns |
| Common wheat | 2015 | 2.1 | a | 33.6 | ns | 39.7 | ns |
| | 2017 | 1.3 | b | 36.1 | ns | 32.9 | ns |
| Durum wheat | 2015 | 2.4 | a | 18.2 | b | 44.6 | ns |
| | 2017 | 1.1 | b | 24.9 | a | 43.3 | ns |
| Oat | 2015 | 2.5 | a | 35.1 | ns | 27.7 | ns |
| | 2017 | 1.8 | b | 33.1 | ns | 29.7 | ns |

Within crop, values followed by different letters are significantly different according to Tukey's test ($p > 0.95$). ns = non-significant.

In common and durum wheat and oat, the increase in the number of kernels was, in turn, due to an increase in the number of heads per plant (62%, 100%, and 38%, respectively).

Conversely, the number of heads per plant was reduced by 44% in barley, although this decline was counterbalanced by the increase in kernels produced per spike (about two-fold) in the first year.

Kernels per spike did not significantly vary in common wheat and oat; on the contrary, they declined in durum wheat (−40%).

Mean kernel weight did not significantly change in the two years, except for durum wheat, which showed 20% heavier kernels in 2015.

Finally, no differences were observed between years in the harvest index for any of the four species (data not shown).

The pattern of nitrogen content in grain was similar to that of grain yield (data not shown), because grain N concentration was not affected by year, and averaged 13 g kg$^{-1}$ in barley, 17 g kg$^{-1}$ in common wheat and oat, and 20 g kg$^{-1}$ in durum wheat.

### 3.3. Effects of Biosolid Application on Biomass and N Content of Winter Cereals at Flowering

In 2017, we carried out an additional sampling at flowering (BBCH 65) to highlight temporal patterns of N uptake by plants, and the probable ensuing N release from fertilizers. Overall, unfertilized plants of all crops showed lower dry weights of all plant parts (VAP, heads, and roots) (Table 3). However, reductions were significant for VAP in barley, common wheat, and oat (−70%, −76%, and −87%, respectively), while in durum wheat, Control was statistically different only from B15 (−79%).

Similarly, barley and oat had meaningfully reduced head biomass (−82 and −89%) without N fertilization, whereas common and durum wheat control plants differed significantly only from B15 (about −87%). Likewise, all unfertilized control plants had reduced root biomass (about −75% for barley and common wheat and about −80% for durum wheat and oat), even if the difference was not significant in barley.

Biosolids and mineral fertilization prompted similar increases compared to control in all parts of plants, as VAP, heads, and roots did not significantly differ among BS and MF treatments in all the cereals.

N concentration of VAP was affected by fertilization only in durum wheat, which had the lowest concentration when unfertilized; moreover, similar values were detected in BS and MF (Table 3).

**Table 3.** Biomass (DW g plant$^{-1}$) and nitrogen concentration (g kg$^{-1}$) of vegetative aboveground plant parts, heads, and roots, as affected by fertilization treatment in barley, common wheat, durum wheat, and oat, at flowering (2017 season).

| Crop | Fertilization | Biomass | | | | | | N Concentration | | | | | |
|---|---|---|---|---|---|---|---|---|---|---|---|---|---|
| | | VAP | | Heads | | Roots | | VAP | | Heads | | Roots | |
| Barley | Control | 0.7 ± 0.2 | b | 0.4 ± 0.2 | b | 0.3 ± 0.1 | ns | 4.2 ± 0.5 | ns | 9.9 ± 0.9 | ns | 11.4 ± 2.3 | b |
| | MF | 2.7 ± 0.3 | a | 2.5 ± 0.1 | a | 1.4 ± 0.6 | ns | 4.5 ± 0.1 | ns | 10.6 ± 0.6 | ns | 8.2 ± 0.5 | b |
| | B5 | 1.8 ± 0.1 | a | 1.8 ± 0.3 | a | 1.1 ± 0.2 | ns | 3.8 ± 0.7 | ns | 10.0 ± 0.2 | ns | 13.5 ± 1.6 | ab |
| | B10 | 2.2 ± 0.2 | a | 2.3 ± 0.0 | a | 0.9 ± 0.2 | ns | 5.0 ± 0.4 | ns | 12.0 ± 0.9 | ns | 16.2 ± 2.4 | ab |
| | B15 | 2.6 ± 0.3 | a | 2.5 ± 0.5 | a | 1.1 ± 0.2 | ns | 5.8 ± 1.2 | ns | 13.4 ± 1.4 | ns | 18.8 ± 2.2 | a |
| Common wheat | Control | 0.7 ± 0.1 | b | 0.1 ± 0.0 | b | 0.4 ± 0.0 | c | 5.8 ± 0.7 | ns | 12.0 ± 1.0 | b | 9.1 ± 1.0 | c |
| | MF | 2.9 ± 0.1 | a | 0.5 ± 0.0 | ab | 1.8 ± 0.0 | a | 7.0 ± 0.2 | ns | 16.8 ± 0.8 | a | 7.2 ± 0.2 | c |
| | B5 | 2.4 ± 0.4 | a | 0.4 ± 0.1 | ab | 1.6 ± 0.1 | ab | 6.7 ± 0.3 | ns | 15.9 ± 1.2 | a | 10.0 ± 0.0 | bc |
| | B10 | 3.3 ± 0.3 | a | 0.6 ± 0.1 | ab | 1.4 ± 0.1 | b | 6.9 ± 0.0 | ns | 15.7 ± 0.0 | a | 12.6 ± 0.2 | ab |
| | B15 | 3.3 ± 0.8 | a | 0.7 ± 0.3 | a | 1.5 ± 0.0 | b | 7.1 ± 0.2 | ns | 16.2 ± 0.9 | a | 13.8 ± 1.4 | a |
| Durum wheat | Control | 0.9 ± 0.3 | b | 0.1 ± 0.0 | b | 0.5 ± 0.2 | b | 5.1 ± 0.3 | c | 14.1 ± 1.5 | b | 6.9 ± 0.2 | c |
| | MF | 3.5 ± 0.2 | ab | 0.7 ± 0.1 | ab | 2.4 ± 0.5 | a | 6.8 ± 0.3 | ab | 15.3 ± 0.0 | a | 5.6 ± 0.1 | c |
| | B5 | 3.1 ± 1.3 | ab | 0.6 ± 0.3 | ab | 2.4 ± 0.5 | a | 6.2 ± 0.0 | b | 14.7 ± 0.5 | ab | 10.8 ± 1.2 | b |
| | B10 | 2.4 ± 0.6 | ab | 0.5 ± 0.0 | ab | 2.2 ± 0.8 | a | 6.7 ± 0.0 | ab | 15.6 ± 0.8 | a | 12.9 ± 0.4 | b |
| | B15 | 4.3 ± 0.8 | a | 0.8 ± 0.2 | a | 2.8 ± 1.7 | a | 7.3 ± 0.0 | a | 18.1 ± 1.1 | a | 23.7 ± 1.0 | a |
| Oat | Control | 0.5 ± 0.1 | b | 0.1 ± 0.0 | b | 0.4 ± 0.1 | c | 5.4 ± 0.6 | ns | 11.9 ± 0.4 | ns | 8.0 ± 0.0 | b |
| | MF | 4.7 ± 0.8 | a | 1.1 ± 0.2 | a | 2.7 ± 0.5 | a | 5.7 ± 0.3 | ns | 9.7 ± 1.1 | ns | 5.9 ± 0.1 | c |
| | B5 | 2.9 ± 0.4 | a | 0.7 ± 0.1 | a | 2.0 ± 0.4 | ab | 5.5 ± 0.1 | ns | 11.8 ± 1.3 | ns | 7.5 ± 0.1 | b |
| | B10 | 3.3 ± 0.7 | a | 0.8 ± 0.1 | a | 1.4 ± 0.1 | bc | 5.7 ± 0.6 | ns | 12.8 ± 2.3 | ns | 12.1 ± 0.0 | a |
| | B15 | 5.0 ± 0.1 | a | 1.1 ± 0.0 | a | 1.8 ± 0.2 | ab | 6.7 ± 0.8 | ns | 12.1 ± 0.6 | ns | 11.7 ± 0.4 | a |

Within crop, values (± standard deviation) followed by different letters are significantly different according to Tukey's test ($p > 0.95$). ns = non-significant.

N concentration of heads in barley and oat was not affected by fertilization treatment, while unfertilized common and durum wheat plants showed lower head N concentration.

Roots of barley and common and durum wheat had the same N concentration in unfertilized and mineral-fertilized pots, and showed an increased concentration when BS were applied. Conversely, oat plants had the lowest N concentration with the mineral fertilization; B5 and C had intermediate levels, while B10 and B15 showed the highest values.

Corresponding to variations in biomass and N concentration, a reduced N content was registered in the different parts of the four cereals, while controls as well as MF and BS were similarly increased (data not shown). For all the crops, the total N content was reduced in unfertilized plants, and those amended with the highest BS rate (B15) also showed the highest nitrogen content (Figure 3). However, plants amended with mineral fertilization and with 5 and 10 Mg ha$^{-1}$ of BS revealed similar values in all the crops.

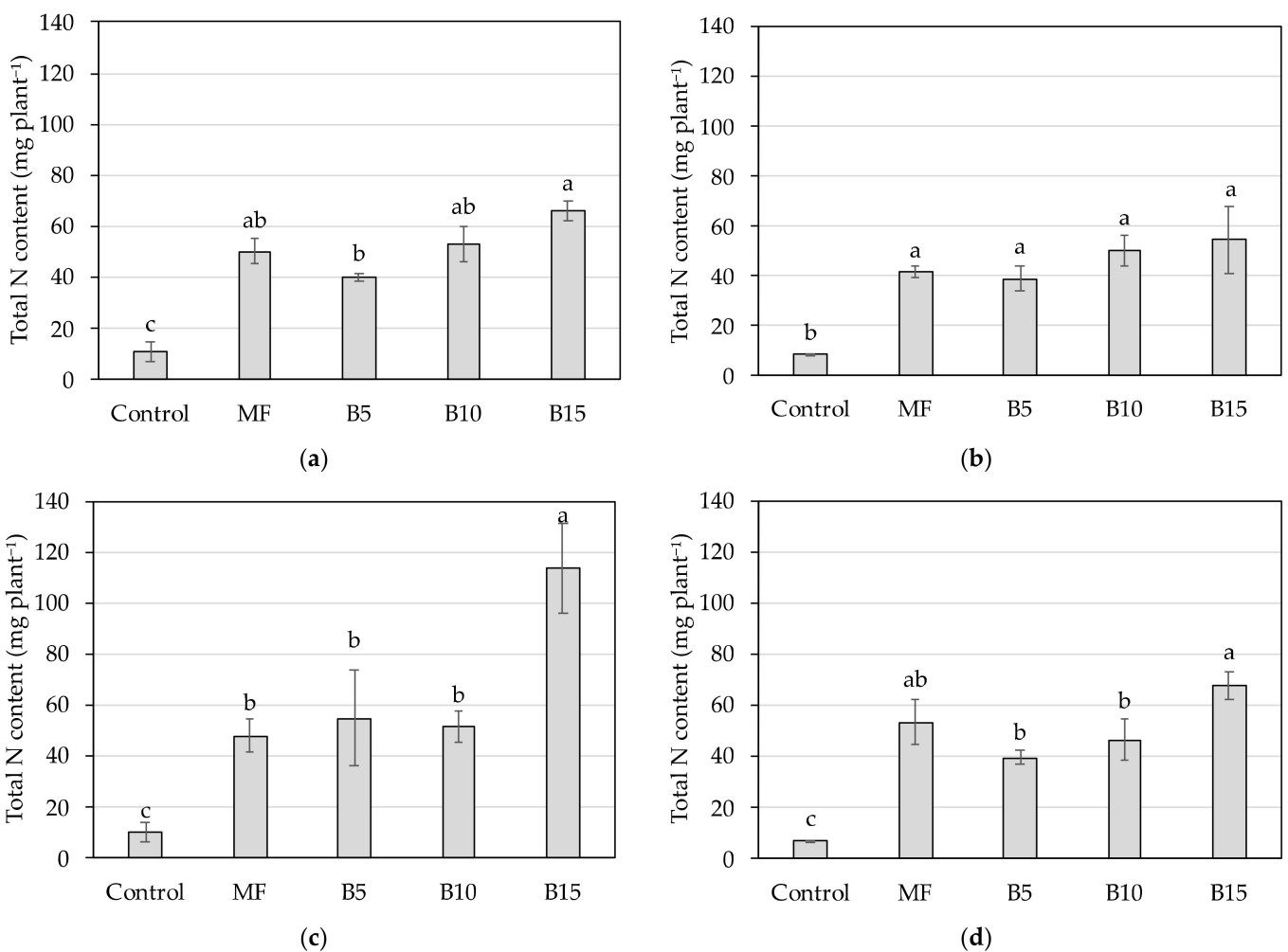

**Figure 3.** Total N content at flowering, as affected by fertilization treatment in: (**a**) barley; (**b**) common wheat; (**c**) durum wheat; and (**d**) oat. Bars with different letters are significantly different according to Tukey's test ($p > 0.95$). Error bars represent standard deviation.

### 3.4. Effects of Biosolid Application on Biomass of Winter Cereals at Maturity

In all cereals, the unfertilized plants showed lower VAP and root biomass compared to the fertilized ones (Figure 4).

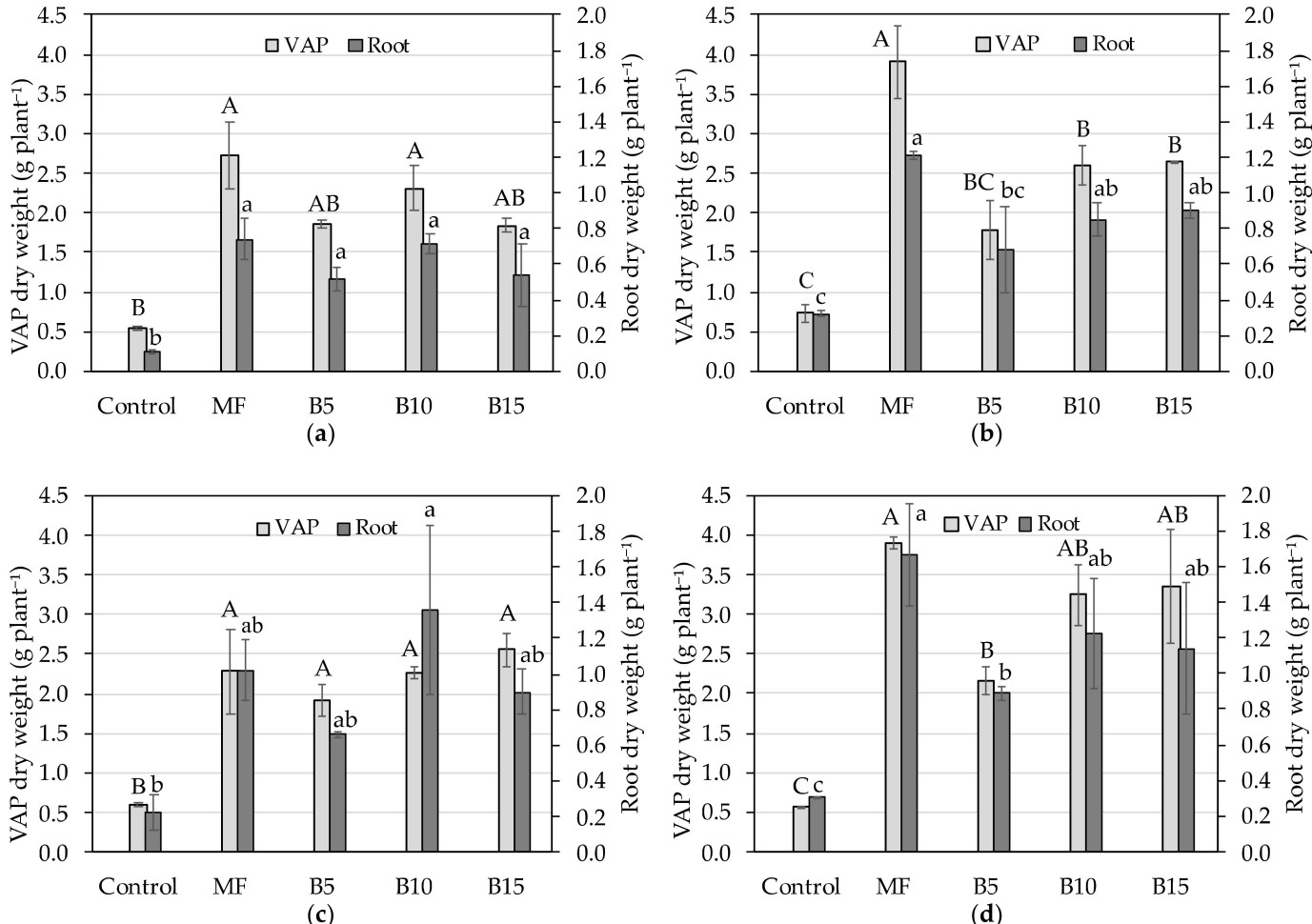

**Figure 4.** Biomass (DW g plant$^{-1}$) of VAP and roots of winter cereals, as affected by fertilizer treatments: (**a**) barley; (**b**) common wheat; (**c**) durum wheat; and (**d**) oat. Bars with different letters are significantly different according to Tukey's test ($p > 0.95$); uppercase letters refer to VAP and lowercase to roots. Error bars represent standard deviation.

Nevertheless, reductions were meaningful for both VAP and roots only for oat ($-82$ and 75%, respectively). Conversely, in barley and common wheat, both VAP and roots were similar in the control plants and B5 fertilized plants; in durum wheat only VAP of fertilized plants statistically differed from the control plants ($-73$%).

However, independently from the applied rate, BS performed similarly to MF in the aerial and root biomass production of barley and durum wheat, while common wheat plants that received BS had 40% and 32% lower VAP and root dry weights than MF. Finally, unfertilized oat significantly differed only from the lowest rate of BS ($-44$% and $-47$% for VAPs and roots, respectively).

### 3.5. Effects of Biosolid Application on Grain Yield and Yield Components of Winter Cereals

Unfertilized crops always produced significantly lower grain yield compared to fertilized plants (about five-fold less for barley, common wheat and oat, and even up to seven for durum wheat) (Figure 5). Additionally, whichever the fertilizer used, crop yield did not differ among MF and BS treatments, with the only exception of common wheat, which showed a lower yield ($-39$%) when fertilized with B5 instead of MF.

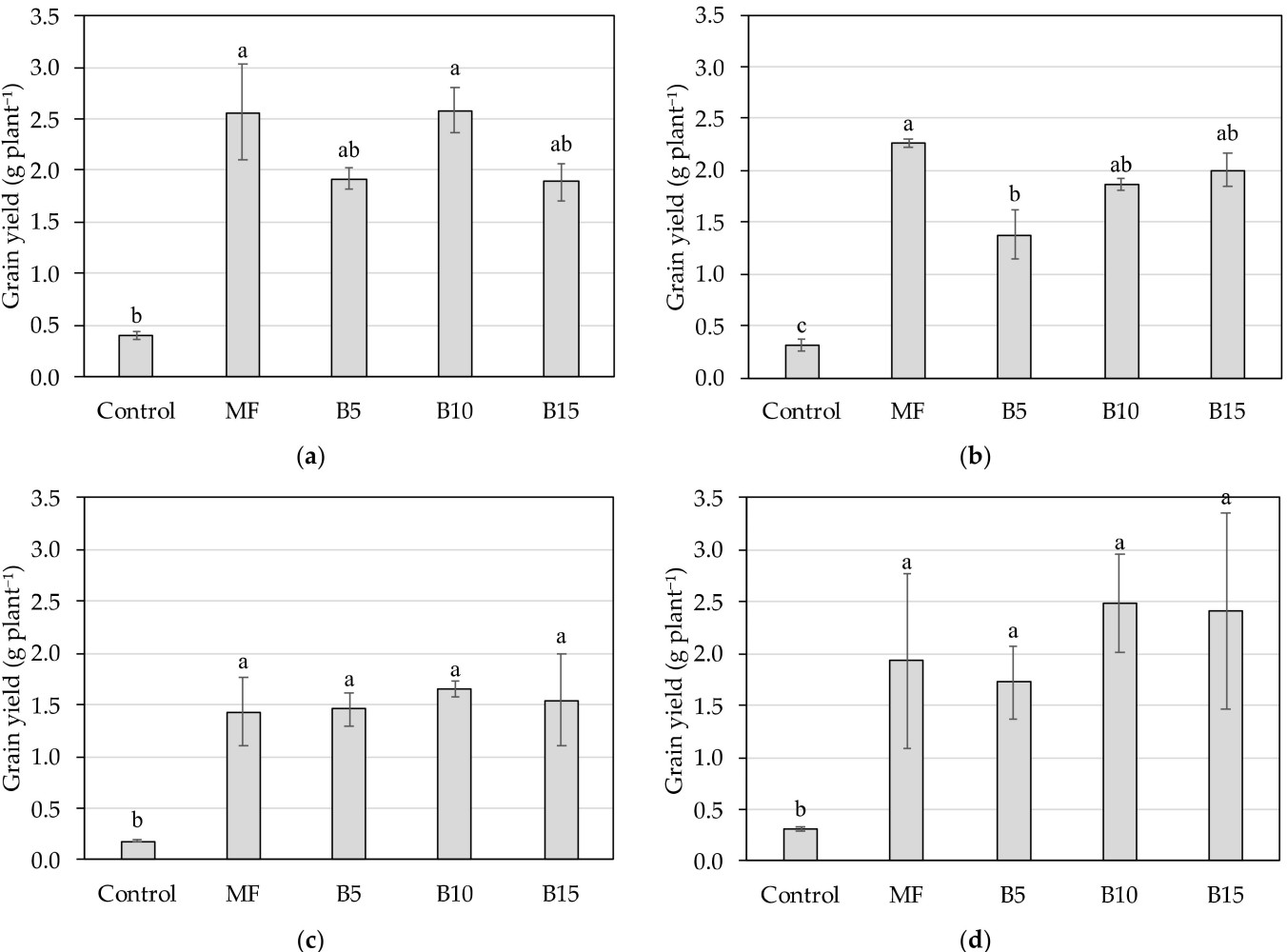

**Figure 5.** Grain yield (DW g plant$^{-1}$) of winter cereals, as affected by fertilizer treatments: (**a**) barley; (**b**) common wheat; (**c**) durum wheat; and (**d**) oat. Bars with different letters are significantly different according to Tukey's test ($p > 0.95$). Error bars represent standard deviation.

In cereals, the yield of an individual plant is the product of: (i) head number per plant; (ii) kernel number per head; and (iii) mean kernel weight.

Heads per plant were unaffected by treatments, except for barley, which had −58% heads in unfertilized plants (Table 4). Likewise, MKW did not differ among fertilization treatments, with the only exception of oat, which showed the lowest MKW with the MF treatment.

As a consequence, the yield increase in fertilized crops was almost entirely derived from the increase in kernels per head than from controls, as this allowed all the crops to produce about 80% more kernels per plant (data not shown). In this regard, both MF and B5 performed similarly in determining the number of kernels per head in barley, common and durum wheat; oat was once again an exception, as it produced 33% less kernels per panicle with the mineral fertilization as compared to the lowest BS rate (B5).

In turn, the boosted number of kernels per head came from a concurrent increase in the number of spikelets and a decrease in the number of florets per spikelet that were terminated in barley and durum wheat. Conversely, in common wheat and oat, the increase was due only to more spikelets, as no significant differences were detected in terminated spikelets (Table 5). Lastly, the head fertility index of control plants was inferior to the fertilized ones in common and durum wheat, while it was similar to MF in control barley and oat, but was increased by BS application at whichever rate (+23 and +17%).

**Table 4.** Yield components of cereals, as affected by fertilization treatment in barley, common wheat, durum wheat, and oat.

| Crop | Fertilization | Heads | | Kernels | | MWK | |
|---|---|---|---|---|---|---|---|
| | | n plant$^{-1}$ | | n head$^{-1}$ | | mg | |
| Barley | Control | 1.1 ± 0.0 | b | 9.0 ± 1.1 | c | 39.9 ± 0.8 | ns |
| | MF | 3.1 ± 0.5 | a | 21.8 ± 4.9 | a | 39.8 ± 0.1 | ns |
| | B5 | 2.2 ± 0.0 | ab | 25.0 ± 4.3 | a | 36.4 ± 7.8 | ns |
| | B10 | 2.9 ± 0.3 | a | 20.7 ± 4.7 | a | 42.9 ± 1.8 | ns |
| | B15 | 2.8 ± 0.6 | a | 16.8 ± 1.1 | b | 40.5 ± 1.9 | ns |
| Common wheat | Control | 1.2 ± 0.1 | b | 8.7 ± 4.2 | b | 30.7 ± 7.0 | ns |
| | MF | 1.9 ± 0.1 | a | 42.9 ± 6.6 | a | 27.6 ± 2.1 | ns |
| | B5 | 1.0 ± 0.0 | b | 40.8 ± 6.8 | a | 34.3 ± 0.9 | ns |
| | B10 | 1.3 ± 0.1 | b | 39.6 ± 7.2 | a | 35.6 ± 3.7 | ns |
| | B15 | 1.3 ± 0.1 | b | 41.4 ± 10.7 | a | 36.5 ± 3.1 | ns |
| Durum wheat | Control | 0.9 ± 0.1 | ns | 5.2 ± 0.2 | c | 37.5 ± 3.2 | ns |
| | MF | 1.3 ± 0.2 | ns | 25.1 ± 0.3 | b | 45.4 ± 3.2 | ns |
| | B5 | 1.2 ± 0.1 | ns | 25.8 ± 0.7 | b | 47.0 ± 0.6 | ns |
| | B10 | 0.9 ± 0.1 | ns | 40.4 ± 2.5 | a | 44.2 ± 5.3 | ns |
| | B15 | 1.4 ± 0.2 | ns | 26.4 ± 3.0 | b | 42.4 ± 0.5 | ns |
| Oat | Control | 1.1 ± 0.1 | ns | 9.0 ± 1.4 | c | 31.4 ± 0.5 | a |
| | MF | 2.9 ± 1.1 | ns | 27.0 ± 9.0 | b | 23.7 ± 1.1 | b |
| | B5 | 1.3 ± 0.0 | ns | 40.2 ± 8.6 | a | 32.2 ± 0.3 | a |
| | B10 | 1.9 ± 0.1 | ns | 41.9 ± 3.3 | a | 31.4 ± 1.5 | a |
| | B15 | 2.3 ± 0.1 | ns | 34.5 ± 5.3 | a | 30.0 ± 3.1 | ab |

Within crop, values (± standard deviation) followed by different letters are significantly different according to Tukey's test ($p > 0.95$). ns = non-significant.

**Table 5.** Harvest index, number of culms, spikelets, and terminated spikelets per plant; head fertility, as affected by fertilization treatment in barley, common wheat, durum wheat, and oat.

| Crop | Fertilization | HI | | Culms | | Spikelets | | Terminated Spikelets | | Head Fertility | |
|---|---|---|---|---|---|---|---|---|---|---|---|
| | | % | | n plant$^{-1}$ | | n head$^{-1}$ | | n head$^{-1}$ | | n g$^{-1}$ | |
| Barley | Control | 43.4 ± 1.2 | b | 1.1 ± 0.0 | b | 14.9 ± 1.8 | b | 5.6 ± 1.1 | a | 128.3 ± 10.4 | b |
| | MF | 48.4 ± 2.8 | ab | 3.5 ± 0.6 | a | 26.5 ± 2.3 | a | 2.3 ± 0.7 | b | 138.5 ± 11.2 | b |
| | B5 | 50.8 ± 0.6 | a | 3.2 ± 0.6 | a | 25.3 ± 2.4 | a | 2.1 ± 0.2 | b | 162.5 ± 10.7 | a |
| | B10 | 52.8 ± 0.9 | a | 3.9 ± 0.8 | a | 23.2 ± 3.4 | ab | 2.0 ± 0.3 | b | 159.1 ± 11.9 | a |
| | B15 | 50.6 ± 1.2 | a | 3.4 ± 1.1 | a | 19.9 ± 1.8 | ab | 2.4 ± 1.2 | ab | 168.4 ± 13.1 | a |
| Common wheat | Control | 29.8 ± 0.7 | c | 1.5 ± 0.3 | b | 12.1 ± 0.7 | b | 4.6 ± 0.0 | ns | 89.6 ± 9.0 | b |
| | MF | 36.6 ± 2.4 | b | 3.5 ± 0.5 | a | 19.8 ± 0.5 | a | 3.6 ± 0.8 | ns | 109.9 ± 9.2 | a |
| | B5 | 43.7 ± 1.1 | a | 3.2 ± 0.7 | a | 20.8 ± 1.1 | a | 3.6 ± 1.3 | ns | 110.5 ± 10.2 | a |
| | B10 | 41.7 ± 1.6 | ab | 3.3 ± 1.0 | a | 20.2 ± 0.1 | a | 3.2 ± 0.0 | ns | 98.7 ± 10.1 | a |
| | B15 | 43.2 ± 2.1 | ab | 2.9 ± 0.5 | a | 20.5 ± 0.4 | a | 3.6 ± 0.4 | ns | 102.7 ± 10.0 | a |
| Durum wheat | Control | 23.4 ± 0.4 | b | 1.0 ± 0.0 | b | 12.5 ± 0.5 | b | 4.1 ± 0.2 | a | 39.3 ± 5.3 | c |
| | MF | 38.5 ± 0.2 | a | 1.7 ± 0.1 | ab | 14.5 ± 0.5 | ab | 1.1 ± 0.1 | bc | 60.8 ± 6.1 | b |
| | B5 | 43.3 ± 0.1 | a | 2.3 ± 0.1 | a | 13.0 ± 1.1 | ab | 1.6 ± 0.2 | b | 74.6 ± 8.9 | a |
| | B10 | 42.2 ± 0.5 | a | 2.2 ± 0.1 | a | 16.5 ± 0.0 | a | 0.7 ± 0.1 | c | 82.2 ± 7.8 | a |
| | B15 | 37.7 ± 4.8 | a | 2.3 ± 0.4 | a | 15.6 ± 1.6 | ab | 1.6 ± 0.1 | b | 80.8 ± 8.8 | a |
| Oat | Control | 35.3 ± 1.8 | ns | 1.5 ± 0.3 | c | 4.9 ± 0.6 | b | 0.0 ± 0.0 | ns | 120.5 ± 11.7 | b |
| | MF | 33.2 ± 9.6 | ns | 4.2 ± 0.0 | a | 16.8 ± 7.1 | ab | 0.6 ± 0.6 | ns | 114.9 ± 10.6 | b |
| | B5 | 44.4 ± 3.1 | ns | 2.4 ± 0.1 | b | 10.4 ± 4.9 | ab | 0.5 ± 0.2 | ns | 146.6 ± 13.1 | a |
| | B10 | 43.4 ± 1.7 | ns | 3.2 ± 0.3 | b | 22.6 ± 1.5 | a | 0.3 ± 0.2 | ns | 135.3 ± 13.9 | a |
| | B15 | 41.9 ± 4.5 | ns | 3.2 ± 0.3 | b | 19.8 ± 4.2 | ab | 0.8 ± 0.5 | ns | 132.3 ± 12.7 | a |

Within crop, values (± standard deviation) followed by different letters are significantly different according to Tukey's test ($p > 0.95$). ns = non-significant.

Likewise, control plants produced 2.2, 1.7, 1.1, and 1.8 less culms per plant (about—66, 54, 52, and 55%, respectively, for barley, common wheat, durum wheat, and oat) (Table 5), but the fertilization treatments did not differ from each other for this character, with the exception of oat, for which the MF increased the number of culms per plant as compared to all the BS treatments (+44%). Similarly, unfertilized plants of barley as well as common

and durum wheat had a lower biomass allocated in grains (e.g., lower HI), while oat did not show significant differences in HI among all the treatments.

### 3.6. Effects of Biosolid Application on N Concentration and N Content of Winter Cereals

In oat plants, the N concentration of all organs was similar among treatments; the same was true for the N concentration of VAP in the other crops and of the roots in durum wheat (Table 6). Contrariwise, common wheat plants amended with B15 had about two-fold higher values of root N concentration than the control plants, while B5 and MF were similar. The N concentration of barley roots was 19% higher with B15 than with MF.

In barley, grain N concentration values did not differ between MF and B5, and were significantly (+25%) increased only by B15. Conversely, the N concentration of grain was higher in the control plants of common and durum wheat and did not differ between MF and B5. The nitrogen content of the VAP of barley and durum wheat was increased in fertilized plants without differences among BS and MF; however, that of common wheat was lower with the single rate of BS than with mineral fertilizers. Root N content was minimum in control plants, and B5 and MF did not significantly differ from each other in barley and durum wheat, while MF was superior in common wheat. In oat, the N contents of VAP and roots were similar among fertilization strategies.

The N contents of grains were lower in control plants and unchanged by different fertilizations in all crops except for common wheat. For this crop, the N contained in grains of plants amended with B5 was lower than that of MF plants or those amended with the double and triple BS rates.

Finally, no differences were found in the NHI of any cereal (data not shown) due to the similar trends detected in the N contents of VAP and grains.

### 3.7. Effects of Biosolid Application on Nitrate Leaching

In 2015, rainfall did not trigger any leaching event, while in the second year, two occurrences happened.

At the first leaching event, on 6 February 2017, drainage water was significantly lower with the application of the three BS rates as compared to that from controls and MF in barley, common wheat, and oat (69, 62, and 41%, respectively), while the decrease was not significant for durum wheat (Table 7). The $NO_3$-N concentration exceeded 50 mg $L^{-1}$ with all BS rates, and was higher than that with the control and MF treatments by about two to four-fold, depending on the species. With B5, the reduction in drainage volume prevailed over the increase of $NO_3$-N concentration in barley, common wheat, and oat, so that the amount of $NO_3$-N leached was comparable with those of controls and MF (Table 7).

At the second percolation event, on 9 March 2017, drainages were somewhat reduced in soils amended with BS in comparison to controls and MF in all crops; however, leachate volume was significantly curtailed only in barley amended with the application of B10 and B15 ($-37\%$). The N concentration in drainage water was significantly increased by all BS treatments and in all cereals, in comparison to MF and controls. As a consequence, no differences were shown in N-$NO_3$ leached with different BS application rates, whose values were comparable to those of MF in barley, common wheat, and durum wheat; however, in oat, BS applications augmented the risk.

Accumulated over the entire crop cycle, drainage water tended to be reduced with the application of whichever BS rate as compared to that from controls and MF; this drop reached significance in barley (Figure 6). In barley, common wheat, and durum wheat, water volume reduction with B5 and B10 balanced higher N concentration; thus, no differences on cumulative N leached were found compared to controls and MF. In oat, total N leached was increased by B10 and B15 application (Figure 7).

**Table 6.** Nitrogen concentration (g kg$^{-1}$) and content (mg plant$^{-1}$) of VAP, roots, and grains, as affected by fertilization treatment in barley, common wheat, durum wheat, and oat.

| Crop | Fertilization | N Concentration | | | | | | N Content | | | | | |
|---|---|---|---|---|---|---|---|---|---|---|---|---|---|
| | | VAP | | Roots | | Grains | | VAP | | Roots | | Grains | |
| Barley | Control | 3.8 ± 0.2 | ns | 14.7 ± 0.7 | ab | 12.0 ± 1.0 | b | 2.0 ± 0.2 | b | 1.6 ± 0.3 | c | 4.9 ± 0.1 | b |
| | MF | 4.5 ± 0.8 | ns | 8.3 ± 0.5 | b | 11.6 ± 0.6 | b | 12.4 ± 1.8 | a | 6.1 ± 1.3 | b | 29.7 ± 10.8 | a |
| | B5 | 4.3 ± 0.2 | ns | 12.5 ± 0.5 | ab | 11.3 ± 0.3 | b | 8.0 ± 0.2 | a | 6.5 ± 0.6 | b | 21.7 ± 0.6 | ab |
| | B10 | 4.8 ± 0.4 | ns | 16.5 ± 2.6 | ab | 13.6 ± 0.7 | ab | 11.0 ± 0.3 | a | 11.9 ± 1.0 | a | 35.2 ± 1.1 | a |
| | B15 | 5.2 ± 0.6 | ns | 17.6 ± 3.9 | a | 14.7 ± 0.1 | a | 9.5 ± 0.4 | a | 9.5 ± 4.1 | a | 27.7 ± 2.8 | a |
| Common wheat | Control | 4.4 ± 0.3 | ns | 5.7 ± 0.4 | c | 19.1 ± 0.1 | a | 3.2 ± 0.6 | c | 1.8 ± 0.1 | c | 5.9 ± 1.2 | c |
| | MF | 3.4 ± 0.2 | ns | 8.4 ± 0.4 | abc | 14.3 ± 0.3 | b | 13.4 ± 0.8 | a | 10.1 ± 0.2 | a | 32.3 ± 0.2 | a |
| | B5 | 4.0 ± 0.1 | ns | 7.7 ± 0.6 | bc | 14.9 ± 0.0 | b | 7.1 ± 1.2 | bc | 5.3 ± 2.3 | bc | 20.5 ± 3.4 | b |
| | B10 | 4.2 ± 0.7 | ns | 11.2 ± 1.9 | ab | 18.4 ± 1.0 | a | 11.1 ± 0.5 | ab | 9.5 ± 0.6 | ab | 34.4 ± 0.9 | a |
| | B15 | 4.3 ± 0.4 | ns | 11.7 ± 0.1 | a | 19.3 ± 0.0 | a | 11.5 ± 1.0 | ab | 10.5 ± 0.4 | a | 38.7 ± 3.0 | a |
| Durum wheat | Control | 4.8 ± 1.0 | ns | 15.9 ± 0.2 | ns | 27.9 ± 3.8 | a | 2.9 ± 0.6 | b | 3.6 ± 1.2 | c | 5.1 ± 0.5 | b |
| | MF | 5.4 ± 0.7 | ns | 9.1 ± 1.7 | ns | 16.5 ± 1.1 | b | 12.3 ± 1.7 | a | 9.3 ± 3.3 | bc | 23.7 ± 3.7 | a |
| | B5 | 6.0 ± 0.0 | ns | 15.0 ± 1.7 | ns | 17.7 ± 3.8 | b | 11.4 ± 1.4 | a | 9.9 ± 0.8 | abc | 25.8 ± 2.9 | a |
| | B10 | 5.6 ± 0.1 | ns | 13.8 ± 4.2 | ns | 16.0 ± 0.1 | b | 12.7 ± 0.2 | a | 18.8 ± 1.0 | ab | 26.4 ± 1.4 | a |
| | B15 | 6.5 ± 0.4 | ns | 17.4 ± 1.5 | ns | 20.4 ± 0.1 | ab | 16.6 ± 2.1 | a | 15.7 ± 3.6 | ab | 31.6 ± 9.0 | a |
| Oat | Control | 6.4 ± 0.2 | ns | 11.3 ± 3.1 | ns | 17.2 ± 1.5 | ns | 3.6 ± 0.3 | ns | 3.4 ± 0.9 | ns | 5.3 ± 0.1 | b |
| | MF | 6.2 ± 3.1 | ns | 6.9 ± 0.8 | ns | 18.3 ± 5.8 | ns | 23.9 ± 13.2 | ns | 11.5 ± 0.7 | ns | 35.3 ± 6.0 | ab |
| | B5 | 5.2 ± 1.5 | ns | 10.7 ± 1.4 | ns | 15.0 ± 0.4 | ns | 11.3 ± 4.0 | ns | 9.6 ± 1.7 | ns | 25.9 ± 5.9 | ab |
| | B10 | 6.0 ± 0.1 | ns | 11.0 ± 1.6 | ns | 15.1 ± 0.2 | ns | 19.5 ± 2.3 | ns | 13.5 ± 5.4 | ns | 37.5 ± 7.7 | ab |
| | B15 | 7.1 ± 0.5 | ns | 11.4 ± 0.6 | ns | 20.8 ± 1.7 | ns | 23.7 ± 2.3 | ns | 13.0 ± 3.6 | ns | 50.2 ± 15.8 | a |

Within crop, values followed by different letters are significantly different according to Tukey's test ($p > 0.95$). ns = non-significant.

**Table 7.** Drainage water, N-NO$_3$ concentration, and N-NO$_3$ leached in the two leaching events (2017), as affected by fertilization treatment in barley, common wheat, durum wheat, and oat.

| Crop | Fertilization | 6 February 2017 | | | | | | 9 March 2017 | | | | | |
|---|---|---|---|---|---|---|---|---|---|---|---|---|---|
| | | Drainage Water | | N-NO$_3$ Concentration | | N-NO$_3$ Leached | | Drainage Water | | N-NO$_3$ Concentration | | N-NO$_3$ Leached | |
| | | mm | | mg dm$^{-3}$ | | kg ha$^{-1}$ | | mm | | mg dm$^{-3}$ | | kg ha$^{-1}$ | |
| Barley | Control | 20.3 ± 4.7 | a | 54.8 ± 15.4 | c | 11.1 ± 4.4 | a | 15.4 ± 4.8 | a | 3.0 ± 1.8 | b | 0.5 ± 0.2 | b |
| | MF | 23.5 ± 9.4 | a | 55.0 ± 20.3 | c | 12.9 ± 4.9 | a | 14.8 ± 5.4 | a | 21.6 ± 7.8 | b | 3.2 ± 1.2 | a |
| | B5 | 4.7 ± 2.5 | b | 145.0 ± 40.8 | b | 6.8 ± 2.7 | b | 13.7 ± 2.9 | a | 78.7 ± 39.7 | a | 10.8 ± 4.8 | a |
| | B10 | 9.2 ± 3.0 | b | 122.8 ± 45.3 | b | 11.3 ± 3.0 | a | 4.5 ± 2.2 | b | 110.9 ± 29.5 | a | 5.0 ± 1.5 | a |
| | B15 | 6.7 ± 2.9 | b | 227.8 ± 70.0 | a | 15.3 ± 4.6 | a | 6.1 ± 2.4 | b | 143.4 ± 47.1 | a | 8.8 ± 2.3 | a |
| Common wheat | Control | 27.4 ± 4.3 | a | 56.7 ± 2.6 | c | 15.5 ± 2.1 | b | 10.4 ± 4.7 | ns | 7.2 ± 2.9 | b | 0.7 ± 0.2 | b |
| | MF | 27.9 ± 10.4 | a | 50.1 ± 12.9 | c | 14.0 ± 4.8 | b | 17.5 ± 5.1 | ns | 28.3 ± 18.0 | b | 5.0 ± 2.3 | ab |
| | B5 | 7.0 ± 4.3 | b | 147.7 ± 37.0 | b | 10.4 ± 3.8 | b | 8.8 ± 3.1 | ns | 110.7 ± 20.4 | a | 9.8 ± 4.1 | ab |
| | B10 | 9.8 ± 5.1 | b | 140.2 ± 41.6 | b | 13.7 ± 5.0 | b | 9.3 ± 3.7 | ns | 148.8 ± 47.6 | a | 13.9 ± 5.8 | a |
| | B15 | 14.7 ± 3.6 | ab | 212.6 ± 57.2 | a | 31.3 ± 10.7 | a | 9.6 ± 2.2 | ns | 113.0 ± 29.4 | a | 10.9 ± 3.8 | ab |
| Durum wheat | Control | 25.0 ± 11.5 | ns | 48.3 ± 6.8 | b | 12.1 ± 4.4 | ns | 18.5 ± 2.0 | ns | 4.3 ± 2.2 | b | 0.8 ± 0.2 | b |
| | MF | 16.4 ± 4.7 | ns | 47.0 ± 13.9 | b | 7.7 ± 3.1 | ns | 17.5 ± 5.1 | ns | 32.1 ± 15.5 | b | 5.6 ± 1.1 | ab |
| | B5 | 10.4 ± 5.7 | ns | 121.3 ± 39.4 | a | 12.6 ± 7.4 | ns | 11.0 ± 8.3 | ns | 107.7 ± 27.2 | a | 11.8 ± 4.0 | ab |
| | B10 | 10.2 ± 8.2 | ns | 138.6 ± 47.1 | a | 14.1 ± 7.2 | ns | 7.6 ± 4.6 | ns | 97.9 ± 27.9 | a | 7.4 ± 3.1 | ab |
| | B15 | 14.8 ± 4.4 | ns | 119.1 ± 45.3 | a | 17.6 ± 11.8 | ns | 15.9 ± 3.6 | ns | 122.4 ± 45.4 | a | 19.4 ± 8.6 | a |
| Oat | Control | 26.3 ± 5.6 | a | 53.9 ± 9.2 | c | 14.2 ± 4.6 | b | 24.5 ± 9.4 | ns | 10.1 ± 4.5 | c | 2.5 ± 0.8 | b |
| | MF | 28.6 ± 8.5 | a | 40.6 ± 8.3 | c | 11.6 ± 2.1 | b | 24.0 ± 10.7 | ns | 16.2 ± 6.1 | c | 3.9 ± 1.2 | b |
| | B5 | 9.2 ± 3.8 | c | 105.6 ± 25.3 | b | 9.7 ± 4.0 | b | 16.4 ± 3.4 | ns | 105.3 ± 38.6 | b | 17.3 ± 2.7 | a |
| | B10 | 19.2 ± 7.4 | b | 154.4 ± 43.3 | ab | 29.6 ± 9.8 | a | 16.9 ± 3.3 | ns | 145.2 ± 23.6 | ab | 24.5 ± 6.8 | a |
| | B15 | 20.2 ± 7.8 | b | 194.5 ± 46.8 | a | 39.2 ± 10.4 | a | 14.3 ± 5.5 | ns | 169.0 ± 31.5 | a | 24.2 ± 9.3 | a |

Within crop, values (±standard deviation) followed by different letters are significantly different according to Tukey's test ($p > 0.95$). ns = non-significant.

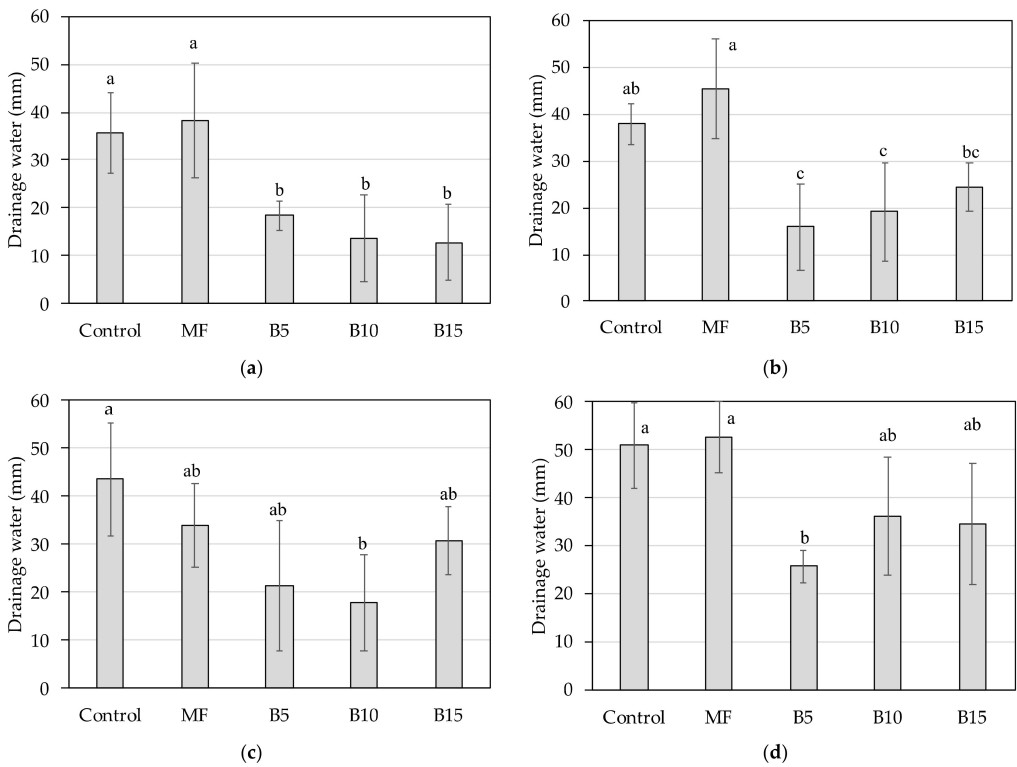

**Figure 6.** Total drainage volumes, as affected by fertilization treatment in (**a**) barley; (**b**) common wheat; (**c**) durum wheat; and (**d**) oat. Bars with different letters are significantly different according to Tukey's test ($p > 0.95$). Error bars represent standard deviation.

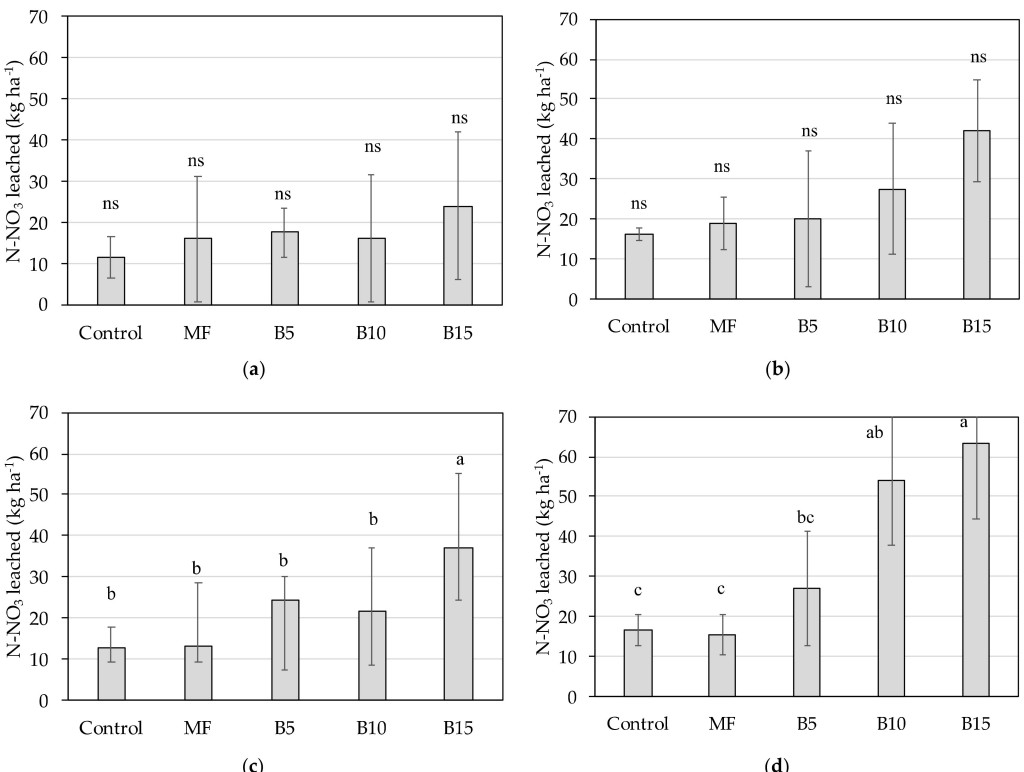

**Figure 7.** Total N-NO$_3$ leached, as affected by fertilization treatment in (**a**) barley; (**b**) common wheat; (**c**) durum wheat; and (**d**) oat. Bars with different letters are significantly different according to Tukey's test ($p > 0.95$). Error bars represent standard deviation.

## 4. Discussion

In the present study, the agronomic and environmental effects of BS applied at three different rates to four of the most cultivated winter cereals in Central Italy were assessed and compared to unfertilized and mineral-fertilized controls, by determining biomass, grain production, and N uptake together with nitrate leaching.

### 4.1. Seasonal Patterns Influenced Biomass and Grain Yield of Winter Cereals

The unsignificant interaction year x fertilization highlighted the fact that the fertilization effect of BS was constant across years. According to [24], even if yearly variation in N uptake by wheat was present, this might not necessarily has been related to changes in biosolid N supply, as other combining effects (mainly water availability) were present. Accordingly, we found that differences in meteorological trends between years influenced the performance of the crops under investigation.

In 2015, all cereals showed higher grain yield because they were able to produce more heads per plants. It is likely that the higher temperatures around sowing increased N mineralization; this augmented the N available in the soil for the crops, prompting better tillering and improving the number of fertile culms per plant, as similarly reported in barley and common wheat by [25], in durum wheat by [26,27], and in oat by [28]. Moreover, in our experiment, N losses through leaching did not occur in the first season, which could have further increased the N available for plants during this initial growth stage.

### 4.2. Biosolids Are Beneficial for Winter Cereals Compared to Unfertilized Controls

When put side by side with the unfertilized controls, biosolids markedly increased biomass production and N uptake both at flowering and maturity, and also the final grain yield of all the cereals. Similarly, analogous increases in biomass, obtained in comparable climatic conditions, were reported in barley [11,29] and in common wheat [30]. In durum wheat, we found that more culms were produced in fertilized plants of durum wheat, corroborating the findings of [31] who observed a positive effect of sewage sludge on the development of tillers in durum wheat due to the nutrient release from BS during the tillering phase. Conversely, the fertile tillering ability of the crop was not improved as the number of spikes was not augmented. Consequently, in the present study, the grain yield of fertilized plants raised from the increase in the number of kernels per spike, confirmed the results of [15] who found that the yield improvement in durum wheat amended with 20, 30, and 40 Mg dry sludge ha$^{-1}$ was due to the rise in kernel number but not to the greater number of spikes produced.

Few studies have been carried out on the effects of BS application on oat production. In forage oats, [32] did not observe significant differences in biomass production among three rates of BS (10, 20, and 30 Mg ha$^{-1}$) and an unfertilized control. The authors hypothesized that the lack of effect was due to sufficient mineral N in the soil before sowing. Conversely, our research was carried out in soil with a low total nitrogen content (0.5 g kg$^{-1}$), which can partly explain the higher biomass of shoots and roots of fertilized plants that we registered. Moreover, the amount of nitrogen required for optimal productivity and responses to N additions have been recognized to be highly dependent on the cultivar, environmental conditions, soil type, and cultivation history [33]; thus, it is also possible that different genotype requirements and efficiency of nitrogen use can explain our different results [34]. Accordingly, [35] also reported that *Avena byzantina* was more responsive to increasing nitrogen rates.

Boosted biomass production was related to the increase in culms formed, as oat almost did not tiller in the unfertilized control, whereas two to four tillers were initiated with the application of fertilizers, as also demonstrated by [36].

Overall, present results corroborated that BS can be an effective source of nutrients for winter cereals even when applied at a low rate [12,37,38]. This was further confirmed by the increase in the N uptake of all crops fertilized with B5 (about 30 mg N plant$^{-1}$ compared to unfertilized ones). We likewise found similar patterns at the sampling carried

out at flowering, thus confirming that BS can sustain N uptake through the entire cropping cycle [11,29].

Nevertheless, besides supplying plants with mineral nutrients, organic amendments can improve soil nutrient availability, microbial activity, and soil physicochemical environment [39]. Thus, it is likely that other factors such as higher water availability and better physical and biological soil conditions could have contributed to the swell in crop growth in our BS treatments. Our previous findings [11] confirm this figure, as they indicated that BS improved soil porosity within a few weeks of application. Similarly, [40] determined that 5 and 15 t ha$^{-1}$ of BS affected fungal communities in the rhizosphere and roots of barley.

### 4.3. Fertilizer Value of Biosolids for Winter Cereals

Even if we did not directly determine the amount of N released by BS during the crop cycle in the present research, we inferred their fertilization potential from the comparison with conventional mineral fertilization.

In barley, durum wheat, and oat, no differences were observed between biosolids and mineral fertilizer in regard to biomass production, both at flowering and maturity, as well as in grain yield; therefore, our results demonstrate that BS can be a reliable option for fertilization of these crops [3,11,15,29], and also confirm the rapid mineralization of BS [6].

The nutrient release pattern of BS can differ from mineral fertilizers because it follows the organic matter mineralization. Yield components of cereals are determined at different times during plant development; thus, their evaluation together with crop characteristics at flowering can be useful for understanding the timing of N movements in the plant-soil-amendment system.

More specifically, in barley, the spike development is indeterminate [41] and the total number of spikelets on the rachis can vary functionally from environmental conditions in the early stages of development. Instead, spikelet abortion occurs later in the season, from a week after the end of new-spikelet initiation up to anthesis, overlapping the period of rapid-stem and rachis elongation [42,43]. We found that BS and MF were similar in determining spikelets per spike and spikelet survival, and both parameters were proven to be affected by the shortage of nutrients [44,45]. Thus, our results suggested that BS, at any application rate, provided an adequate amount of N for barley, somehow comparable to that from the mineral fertilization, both at pre- and post-steam erection stages (30 and 120 kg N ha$^{-1}$, respectively). This was further confirmed by the increase that BS prompted in the head fertility index, as it is primarily determined during the stem elongation period (i.e., the time between terminal spikelet phase and anthesis) [46]. Moreover, we can deduce a slow release from BS throughout the entire crop cycle because N concentrations of VAP at flowering were similar in BS and MF. However, root N concentrations in BS treatments, both at flowering and maturity, were improved together with the N concentration in the grain. This may indicate that N availability in BS-amended plants was consistent during the entire crop cycle, and that the root system was actively uptaking N from the soil even during grain filling and in the late development phases.

We did not find differences between BS and MF in determining the grain yield of durum wheat, contrary to [16] who found an increase with different rates (20, 50, and 100 Mg ha$^{-1}$) of BS as compared to the mineral fertilization. However, only 35 kg ha$^{-1}$ of N were added as urea in their research [16]. Accordingly, variances in N availability from mineral and BS fertilizations may be responsible for the differences they found among treatments. They also highlighted a linear increase in the aboveground biomass of durum wheat, with increasing doses of sewage sludge; on the other hand, we did not find differences among BS rates. It is likely that the later sowing we performed did not allow the crop to fully benefit from the higher N availability associated with higher BS rates due to a shortened vegetative phase [47,48].

To the best of our knowledge, no research has evaluated how BS can affect grain yield in oat. Recently, [49] reported parallel responses in yield, weight of 1000 seeds, and physiological seed quality of oat, with both mineral and organic fertilization (vermi-

compost). In the present research, we found that oat plants produced higher biomass with MF compared to B5, mainly depending on the higher number of culms, but enhanced tillering did not bring differences in the number of panicles between the two fertilization treatments. Similarly, [32] obtained an increase in forage yield with BS rates of 500, 700, and 1000 $m^3$ $ha^{-1}$, but no differences were detected between BS and mineral fertilizer application. The lack of difference in grain yield among the BS rates we found is consistent with [50] who demonstrated that oat positively responded to nitrogen fertilization in terms of plant growth, although high N rates had limited effects on final grain yield. Generally, oat has proven to be highly responsive to increasing fertilizer N at low rates, but less responsive to higher N rates [51]. Previous studies observed optimal N responsiveness in oat in the range of 30–120 kg N $ha^{-1}$ [52,53]; however, there were small grain yield increases when the N rate was increased to 120 kg $ha^{-1}$ [54]. We applied 150 kg $ha^{-1}$ of N with the MF, which could explain why the BS application prompted only slight and not significant increases in oat grain yield as compared to the standard fertilization. In our study, having no differences in the number of spikelets per panicle and spikelet abortion indicates that similar PAN between treatments was present in the soil during panicle initiation and anthesis. According to [55], adequate N availability can produce a significantly improved floret set and reduced floret abortion, resulting in an increased number of kernels per panicle. Interestingly, oat was the only crop within the present study that exhibited differences in mean kernel weight, which is the last determined yield component in cereals. As established by [55], the increased yield potential with the application of fertilizer N was associated with enhanced grain-filling rates, the number of spikelets per panicle, and mean grain weight. Thus, in our research, the upsurge in the mean kernel weight of oat plants receiving BS was probably driven by higher N uptake during grain filling or by an increased remobilization of pre-anthesis resources. This could mean that the enduring mineralization of BS allowed for a significant amount of N to be supplied to oat plants, which subsequently ensured both a gradual growth in the number of grain per panicle and an increase in MKW.

Contrary to the other three crops, the present results indicate that for common wheat, the BS amount that can be applied annually to agricultural soils in Italy (5 Mg DM $ha^{-1}$, viz. B5) was not as effective as MF for crop growth. Differences between MF and BS in root and vegetative aboveground part biomass, and consequently in the N uptake, were observed both at flowering and maturity, pointing out that during the entire crop cycle, BS at 5 Mg $ha^{-1}$ did not provide an optimal amount of N for common wheat. Moreover, the double and triple BS rate showed similar fertilizer values to MF yields but were also related to an increased risk of N leaching, particularly if intense rainfall occurred in late winter and before the crop had sufficient ETP and N uptake [11]. Likewise, the grain yield of common wheat declined in B5 as compared to MF. The main components of wheat yield are the number and weight of kernels [56]. In turn, the number of kernels results from the number of heads per plant and the number of kernels per head, and the critical period for determination of these yield components in wheat is during the spike-growing period (from terminal spikelet to anthesis) [57]. Conversely, the weight of kernels is determined by the rate and duration of the grain-filling phase, and thus reflects post-anthesis conditions [58].

Within the present research, the lower grain yield was due to less heads per plant, while MKW was not affected. The number of culms per plant was similar between MF and BS, although the latter had less productive tillers; thus, we maintained that differences in the N supply between the two fertilizers were present in the period of tiller mortality that occurs from jointing to anthesis [59]. On the contrary, probably similar N was available for uptake during grain filling due to the top-dressing N mineral fertilization and from the BS release. This could further be evidenced by the lack of response from common wheat to the two types of fertilization in terms of survival of florets, and consequently in the number of kernels per head. Accordingly, the lower N content in the grains of the single BS rate compared to those of MF could be explained by less N remobilization to grains due to less N content at anthesis [60].

Grain yield component development in common wheat and durum wheat is similar; accordingly, the different results we reported in the yield performances of the two species may likely be explained by the higher N demand of common wheat [61].

Finally, the differences in crop response we found could be ascribed to different patterns of N release from the two types of fertilizer used, which influenced biomass accumulation, grain yield, and N uptake, as also reported by other authors [11,30,37]. However, it should be underlined that even if the majority of mineralizable N is released rapidly in biosolid-amended soil in the year of application [6], further research should be performed in order to investigate the N fertilizer equivalency of BS under multi-application conditions, since each yearly rate can deliver residual organic N in soil, capable of releasing PAN for years after application [29].

### 4.4. Type and Rate of Fertilizers Can Affect Nitrate Leaching Risk

N leaching is the main pathway of N loss in the Mediterranean regions, and it is also regarded as the greatest environmental question to be solved for BS application to arable soils [62]. When BS are distributed to the soil in winter, as in the present research, leaching risk is most likely to occur in early spring, when elevated drainage volumes are associated with still limited crop needs [11]. In our research, rainfall was limited and lower than the long-term average in both cropping seasons. Water percolation and associated N leaching occurred only in one out of the two years, but it is a usual condition in Mediterranean soils where N accumulates and is then leached out of the soil profile, when heavy rainfall occurs [63]. Similarly, [64] demonstrated that the $NO_3$-N content in the 0–60 cm soil layer was the highest in spring, subsequently decreasing with crop growth.

In 2015 rainfall did not trigger any leaching event because rainfall at the end of January up to the beginning of February came after a period without precipitation and consequently, soil field capacity was able to receive water without percolation. This resulted in more N remaining within the upper soil profile and available for plant uptake, contributing to an increase in the final grain yield. In the second year of experimentation, two leaching events occurred and we found reduced drainage volumes in the BS-amended soil as compared to controls and MF. This can primarily be ascribed to the high organic matter content of BS that modifies soil porosity, reduces bulk density, and improves aggregate stability and water-holding capacity [65]. Previously, BS have been reported to improve the soil structure and water holding capacity [66–68], likewise in Italian sandy-loam and coarse-textured soils by [69] and [11], respectively.

On the contrary, the $N$-$NO_3$ concentrations in drainages were increased by BS application compared to controls and MF in both leaching events and in all the four species. As leachates were collected 52 and 82 days after BS application, our results highlighted the presence of noteworthy mineralizable nitrogen content, which is believed to be about 14% of total N in dewatered digested sludge [6].

However, we revealed that the higher N concentrations in the leachates of BS-amended soils were balanced by the lower water volumes, so that the amount of N leached was similar to when MF and BS were applied at 5 and 10 Mg ha$^{-1}$ rates and consistently increased only with BS application of 15 Mg ha$^{-1}$. Our results thus confirmed that the increased amount of nitrate that may move below the root zone is the primary drawback to the increase of BS application rates, as also stated by [70].

Our findings additionally underlined the fact that the application of a single high rate in three years (B15) is prone to increased risks of leaching as compared to a yearly application (B5), which in turn was proved to be not upsetting and certainly comparable to conventional mineral fertilization, as already reported by previous research [11,70–72].

As well as application rates, we also found different behaviors in the four studied winter cereals in determining N leaching. Overall, the N-leaching risk increased in the following order: barley < common wheat = durum wheat < oat. We can infer that the different development rates of the four cereals under the Mediterranean climate [73,74]

could have determined the dissimilarities in water uptake by the crops; however, lack of data at early development stages did not allow more precise assumptions.

**5. Conclusions**

In the present research, we showed that seasonal differences in temperature and rainfall conditions following BS application can exert significant impact both on N availability for the crop and on N leaching risks because they affected the pattern of N release and its drainage.

Moreover, we hypothesized that the amount of mineral N continuously released throughout the growth cycle by BS could differently affect the plant growth and grain yield of diverse cereals. Our results confirm this hypothesis, demonstrating that whether 5 Mg ha$^{-1}$ of biosolids could replace mineral fertilization depended on the particular cereal. In barley, durum wheat, and oat, this rate supplied sufficient available N to sustain yields, equaling the productivity of mineral-fertilized plants. Conversely, in common wheat, results comparable to MF were achieved only by doubling the rate of BS application (10 Mg ha$^{-1}$). The risk of N leaching associated with the application of 5 Mg ha$^{-1}$ of biosolids was similar to that of mineral fertilization, confirming that roots promptly accumulate the N progressively mineralized by the BS.

Thus, biosolids can effectively replace mineral N as fertilizers in the cultivation of winter cereals; however, when land application systems are designed (soil and climate of the area, and accordingly, timing), application rates must be determined specifically for the crop to which biosolids are going to be applied.

**Supplementary Materials:** The following is available online at https://www.mdpi.com/article/10.3390/agronomy11081482/s1. Table S1: Nitrogen concentration (g kg$^{-1}$) and content (mg plant$^{-1}$) of vegetative aboveground parts (VAP) and roots as affected by year treatment in barley, common wheat, durum wheat and oat.

**Author Contributions:** Conceptualization, methodology, validation, writing—review and editing, S.P.; formal analysis, investigation, S.P. and A.R.; writing—original draft preparation, S.P. and I.A. All authors have read and agreed to the published version of the manuscript.

**Funding:** This research received no external funding.

**Data Availability Statement:** The data presented in this study are available on request from the corresponding author.

**Conflicts of Interest:** The authors declare no conflict of interest.

**Abbreviations**

BS: Biosolids; MF: mineral fertilization; C: Control; N: Nitrogen; PAN: Plant Available Nitrogen; DM: Dry Matter; DW: Dry Weight; HI: Harvest Index; NHI: Nitrogen Harvest Index; VAP: Vegetative Aboveground Parts; MKW: Mean Kernel Weight.

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
