# Peer review of "Biosolids Benefit Yield and Nitrogen Uptake in Winter Cereals without Excess Risk of N Leaching"

_agronomy, doi:10.3390/agronomy11081482_

Round 1
Reviewer 1 Report
The submitted manuscript examines the effect of biosolids on four cereals at doses determined in accordance with Italian legislation. The article examines numerous plant parameters and the risk of nitrogen leaching in two growing seasons. The presentation of the work is detailed, precise and may be of interest to a wider audience due to the large number of parameters examined.
Comments:
Figure 1: a legend or description for white and black dots is necessary.
For more precise data presentation SD values should be included in the tables and figures.
Line 655-660 is gives redundant information with results presentation.
The conclusions should focus on the results of the present experiment. General statements like line 691-694 can be omitted.
Author Response
Dear Reviewer 1,
We are very grateful for your valuable comments - the changes you suggested have undoubtedly improved our manuscript.
We have addressed all the concerns you raised, providing a point-by-point answer on how we handled each suggestion. Our reply (R) is in italics right after each comment of yours (C). Changes made to the text have been highlighted.
Sincerely,
Point-by-point answer to Reviewer 1:
Comment: The submitted manuscript examines the effect of biosolids on four cereals at doses determined in accordance with Italian legislation. The article examines numerous plant parameters and the risk of nitrogen leaching in two growing seasons. The presentation of the work is detailed, precise and may be of interest to a wider audience due to the large number of parameters examined.
Reply: Thank you for your valuable comments.
C: Figure 1: a legend or description for white and black dots is necessary.
R: Done.
C: For more precise data presentation SD values should be included in the tables and figures.
R: Done.
C: Line 655-660 is gives redundant information with results presentation.
R: We amended the text and omitted the calling back of results.
C: The conclusions should focus on the results of the present experiment. General statements like line 691-694 can be omitted.
R: According to your suggestion we have improved this section in the revised manuscript, deleting those statements and focusing more on our main results about yield and N leaching as affected by BS and BS rates. The changes in the file have been evidenced. Thank you for showing it was not well-defined.
Reviewer 2 Report
Review Manuscript ID Agronomy 1292254
Biosolids benefit yield and nitrogen uptake in winter cereals 3 without excess risk of N leaching.
To Authors
This is not very well organized and prepared paper. I found this manuscript interesting and partly innovative, but a few questions must be explained. The best part of the “ms” seems to be Discussion.
Critical review:
- The Abstract and Introduction as they stand are somewhat unreadable. Lots of information, but I have the impression that the results were already known. How do you explain that?
- I do not think that the tables presented in this way are acceptable. Change to Figures.
- The conclusions are unacceptable. They should be clear and transparent, short and not descriptive. Necessarily to change.
- “Agronomy” (journal) is very prestigious one. Personally I read papers from this journal systematically. In this context try to do your best.
Author Response
Dear Reviewer 2,
We are very grateful for your valuable comments - the changes you suggested have undoubtedly improved our manuscript.
We have addressed all the concerns you raised, providing a point-by-point answer on how we handled each suggestion. Our reply (R) is in italics right after each comment of yours (C). Changes made to the text have been highlighted.
Sincerely,
Point-by-point answer to Reviewer 2:
Comment: The Abstract and Introduction as they stand are somewhat unreadable. Lots of information, but I have the impression that the results were already known. How do you explain that?
Reply: We revised the abstract, focusing more on our specific results about the effects on grain yield and N leaching of BS rates, as suggested. We have amended also the Introduction correspondingly to your comments. We omitted the literature evidence not relevant to highlight the research gaps and the significance of our research.
C: I do not think that the tables presented in this way are acceptable. Change to Figures.
R: Thank you for your suggestion, we graphed the results of VAP and grain dry weight (from table 2 and 3). We will not show the statistically not significant data of N concentration, but we prepared a supplemental table with the N content results that could be of some interest for the readership.
C: The conclusions are unacceptable. They should be clear and transparent, short and not descriptive. Necessarily to change.
R: According to your suggestion we have improved this section in the revised manuscript, deleting redundant and general statements and focusing more on our main results about yield and N leaching as affected by BS and BS rates. The changes in the file have been evidenced. Thank you for showing it was not well-defined.
C: “Agronomy” (journal) is very prestigious one. Personally I read papers from this journal systematically. In this context try to do your best.
R: We agree, so we did our best to improve the manuscript.
